# Clustering of fast gyrotactic particles in low-Reynolds-number flow

**Jenny Lynn Ongue Almerol** [1]*, **Marissa Pastor Liponhay** [2]

**1** Department of Physics, University of San Carlos, Cebu City, Cebu, Philippines, **2** Analytics, Computing, and Complex Systems laboratory (ACCeSs@AIM), Asian Insitute of Management, Makati City, Philippines

\* jloalmerol@usc.edu.ph

**Data Availability Statement:** All relevant data are within the paper.

**Funding:** JA and ML acknowledge the Department of Science and Technology (DOST)-SEI Accelerated Science and Technology Human

## Abstract

Systems of particles in turbulent flows exhibit clustering where particles form patches in certain regions of space. Previous studies have shown that motile particles accumulate inside the vortices and in downwelling regions, while light and heavy non-motile particles accumulate inside and outside the vortices, respectively. While strong clustering is generated in regions of high vorticity, clustering of motile particles is still observed in fluid flows where vortices are short-lived. In this study, we investigate the clustering of fast swimming particles in a low-Reynolds-number turbulent flow and characterize the probability distributions of particle speed and acceleration and their influence on particle clustering. We simulate gyrotactic swimming particles in a cubic system with homogeneous and isotropic turbulent flow. Here, the swimming velocity explored is relatively faster than what has been explored in other reports. The fluid flow is produced by conducting a direct numerical simulation of the Navier-Stokes equation. In contrast with the previous results, our results show that swimming particles can accumulate outside the vortices, and clustering is dictated by the swimming number and is invariant with the stability number. We have also found that highly clustered particles are sufficiently characterized by their acceleration, where the increase in the acceleration frequency distribution of the most clustered particles suggests a direct influence of acceleration on clustering. Furthermore, the acceleration of the most clustered particles resides in acceleration values where a cross-over in the acceleration PDFs are observed, an indicator that particle acceleration generates clustering. Our findings on motile particles clustering can be applied to understanding the behavior of faster natural or artificial swimmers.

## Introduction

Systems of particles in turbulent flows are widely observed in many natural phenomena such as dust/sand storms, clouds, bacterial suspensions, and in oceans, lakes, and reservoirs [1, 2]. Aside from their ubiquitous nature, these particle systems have significant impacts on public health and safety. Thus, understanding their dynamics is relevant and provides insights into these phenomena. For example, understanding the dynamics of dispersed particles in turbulent flows is essential in predicting precipitation and the occurrence of dust storms [3, 4].

Resource Development Program (ASTHRDP) and the University of San Carlos for supporting this research. ML acknowledges the Department of Science and Technology (DOST) of the Philippines with Project No. 8419, 2020 under the Collaborative R&D to Leverage the Economy (CRADLE) Program (https://s4cp.dost.gov.ph/programs/cradle/). The funders had no role in study design, data collection, and analysis, decision to publish, or preparation of the manuscript.

**Competing interests:** The authors have declared that no competing interests exist.

Modeling the migration of motile microorganisms is helpful in the ecological risk assessments associated with harmful algae in oceans and estuaries [5, 6] and in predicting their interaction with other particles that lead to ecological harm such as disease transmission [7]. Application of the insights from these studies can be extended to issues concerning human society such as contamination of urban water systems, which is a significant source of disease outbreaks, affecting a few to thousands in a single instance of an outbreak [8].

While these works focused on the migration of systems of particles, another interesting behavior of particles in turbulent flows is that particles accumulate and form patches in certain regions of space, i.e., particles do not remain uniformly distributed [1]. For example, photo-synthetic motile algae or gyrotactic phytoplanktons, which swim toward the upper part of the ocean with the brightest light, exhibit clustering wherein motile phytoplanktons are observed to be more clustered than non-motile ones [9, 10]. Mathematical models and numerical simulations have shown that this complex behavior emerges from the coupling of motility and shear in the vortical flow [9–11]. Previous studies have shown that gyrotactic cells form clusters inside the vortices and in downwelling regions or regions with downward flow [9, 11, 12]. Similar behavior is observed for light non-motile particles in which particles preferentially concentrate inside the regions of high vorticity. In contrast, heavy non-motile particles accumulate outside these vortices, which act as centrifuges ejecting the particles [13–15].

The presence of these vortices is correlated with intense and persistent centripetal particle acceleration [16]. Fluid acceleration in high-vorticity regions generates stronger multifractal cell clustering [10]. However, clusters of phytoplankton cells are still observed in turbulent flows where vortices are short-lived [9, 11]. In these low-Reynolds-number turbulent flows, non-motile cells are observed to be well-distributed, and the clustering of motile cells is dependent on the swimming parameters such as speed and stability [9, 11]. Moreover, clustering only occurs from certain combinations of swimming velocity and stability, which results in either clustering inside the vortices or downwelling regions. In this study, we investigate the clustering of fast swimming gyrotactic particles embedded in a low-Reynolds-number turbulent flow and focus on characterizing the particle speed and acceleration and its influence on particle clustering.

We simulate dimensionless swimming particles in a low-Reynolds-number turbulent flow, exploring combinations of swimming parameters (swimming speed and stability) that generates particle clustering. Here we consider swimming speeds that are relatively high compared to the previous studies [9, 11, 17]. In the next sections, we describe the methods for the numerical simulations and the numerical procedure for the data collection. Furthermore, we also discuss the clustering of particles investigated by measuring entropy [18, 19]. We also present the speed and acceleration distributions of particles for different combinations of swimming parameters.

## Methods

The system consists of gyrotactic swimming particles embedded in a cube with low-Reynolds-number homogeneous isotropic turbulent flow. In this section, we first discuss the background of gyrotactic swimming particles, then describe the numerical simulation of the turbulent flow embedded with swimming particles. Lastly, we discuss the numerical procedure for data collection.

### Gyrotaxis model

We investigate the clustering of particles and characterize the probability distributions of their speed and acceleration by allowing the particles that are seeded uniformly throughout the cube

to disperse until their mass distribution reaches a statistically steady state. Each particle moves with a net velocity that is the sum of its intrinsic velocity and the velocity of the fluid affecting it. The fluid velocity is taken from the direct numerical simulation (DNS) of the Navier-Stokes (NS) equation using a dealiased pseudo-spectral code [20, 21], while the intrinsic swimming velocity is based on the gyrotaxis model of swimming [9, 10]. In particular, the swimming particle follows the equations of motion given by

$$\mathbf{v} = \mathbf{u}(\mathbf{x}, t) + v_s \mathbf{p} \tag{1}$$

$$\frac{d\mathbf{p}}{dt} = \frac{1}{2B}[\hat{k} - (\hat{k} \cdot \mathbf{p})\mathbf{p}] + \frac{1}{2}(\boldsymbol{\omega} \times \mathbf{p}) \tag{2}$$

where $\mathbf{v}$ is the net velocity of the particle, $\boldsymbol{u}$ is the fluid velocity, $\boldsymbol{x}$ is the position, $t$ is time, $v_s$ is the swimming speed, $\boldsymbol{p}$ is the swimming orientation, $B$ is the reorientation time, $\hat{k}$ is the unit vector in the $z$-direction, and $\boldsymbol{\omega} = \nabla \times \mathbf{u}$ is the vorticity [9, 10]. The first term of Eq 1 describes the tendency of the particle to remain aligned along the vertical direction, while the second term captures the tendency of vorticity to overturn the particle by imposing a viscous torque on it [9, 10]. Without swimming, the particles are considered as tracer particles as the second term of Eq 1 becomes zero, and the particle becomes affected only by the fluid velocity.

## Numerical simulation

To simulate the system, we first create a three-dimensional (3D) cube with a turbulent fluid flow. Then, we embed tracer particles and incorporate the swimming mechanism to the particles as described in Eqs 1 and 2. The numerical implementation is written in a parallelized Python code, where the solver for the DNS of the NS equation from references [20, 21] is used. DNS is a valuable tool in fluid dynamics that uses methods of high order and accuracy and has been used to simulate homogeneous isotropic turbulence [22]. Here we use the Python NS solver as it is easy to modify or extend and, at the same time, still provides accurate and reliable data.

The simulation process is divided into two parts, (a) the fluid part calculation and (b) the particle part calculation (as shown in Fig 1). The fluid part includes the computation of fluid velocity and fluid vorticity using DNS, while the particle part includes the computation of the position, velocity, and swimming orientation of the particles. The particle positions are initialized and allocated among each CPU for the parallel implementation. The fluid velocity and vorticity at the locations of particles are calculated using the tricubic interpolation method [23]. Eq 2, on the other hand, is numerically solved using the fourth-order Runge-Kutta method [24]. As for the time-stepping scheme for the particle calculation part, the Adams-Bashforth 3/ Moulton 4 step predictor/corrector (ABM) method is used [25].

## Numerical procedure

We used a periodic cube with a length of $L = [2\pi, 2\pi, 2\pi]$ and a resolution of $96^3$ grid points, which ensures accurate resolution at small scales. The turbulent fluid flow is at Reynolds number $Re_\lambda = 59$, a turbulent fluid flow where tracer particles are observed to be uniformly distributed [9]. This guarantees that no clustering of particles is induced by the fluid vorticity alone. The time step used is $\Delta t = 0.002$, which satisfies the Courant-Friedrichs-Lewy criterion to resolve temporal scales in the DNS of NS equation. When a statistically steady state is achieved ($\sim 10 - 15$ eddy turnover times [22]), each particle is tracked, and its position, velocity, and acceleration per time step are recorded for the next 500 time steps. The numerical simulations

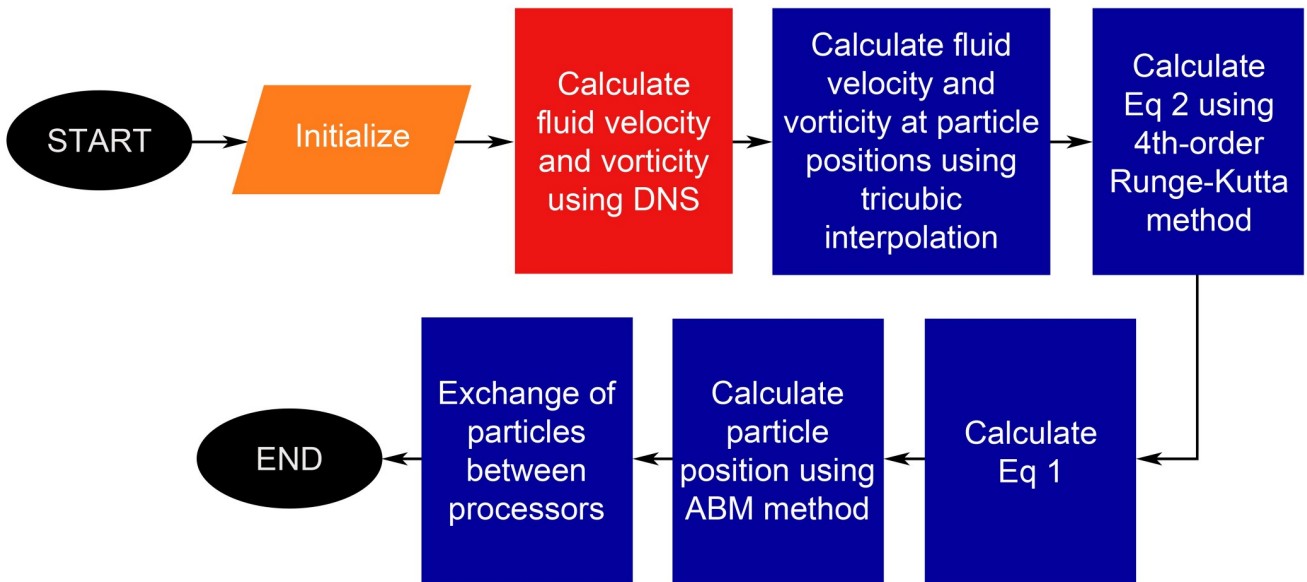

**Fig 1. Numerical simulation workflow.** The red box corresponds to the fluid calculation while the blue boxes correspond to the particle calculations.

are performed on a 40-CPU Supermicro X11DAi-N supercomputer, and each run takes $\sim$ 2.8 hours for tracer particles and $\sim$ 4.1 to 5.2 hours for swimming particles.

Swimming parameter values explored are $v_s \in [0 : 30]$ and $B \in [1 : 50]$. Following the standard quantification of parameters used in previous studies, we de-dimensionalize the parameters with the Kolmogorov scales so that $\Phi = v_s/u_\eta$ and $\Psi = B/\tau_\eta$, where $u_\eta = (\epsilon v)^{1/4}$ and $\tau_\eta = (\epsilon/v)^{1/2}$. Parameters $\Phi$ and $\Psi$ are the swimming number and the stability number, respectively. Here, both ranges of dimensionless parameters selected are greater relative to the previous studies [9, 11], which have not been explored. This is also to investigate whether particle acceleration alone can produce clustering. We extend our simulations using other Reynolds numbers $Re_\lambda \in [21, 36]$, where clustering of gyrotactic cells was previously observed [9, 11, 17], to confirm whether clustering of fast swimming particles will also be observed in such flows. The parameters used in this study are summarized in S1 Table.

The post-processing of data includes the measurement of the clustering of particles and the characterization of the velocity and acceleration of the particles. The two ways with which the clustering is measured are by calculating (a) particle density and (b) entropy. To measure the particle density, the cube is divided into smaller cubes or bins of length $r$. The number of particles contained in each bin is counted, then the particle density is calculated using the equation $\rho = nL^3/(Nr^3)$, where $n$ is the number of particles per bin and $N$ is the total number of particles normalized by $r^3$. Using the bins created, we calculate the entropy $H = -\sum_i p_i \log (p_i/r^3) - \sum_i p_i \log r^3$, where $p_i$ is the probability per bin [18, 19]. The maximum entropy $H_{max} = \ln(N) = 11.53$ corresponds to uniformly distributed particles. Thus, high particle clustering corresponds to a low entropy value. The characterization of the velocity and acceleration, on the other hand, is done by taking their probability distributions (PDF). We also explore the distribution of the longitudinal and centripetal acceleration, $a_L \equiv (a \cdot \hat{v})\hat{v}$ and $a_C \equiv a \times \hat{v}$, where $\hat{v}$ is the particle velocity unit vector [16].

## Results and discussions

In this section, we present the results when $Re_\lambda = 59$, a turbulent fluid flow where tracer particles are observed to be uniformly distributed. The same experiments have been done for other

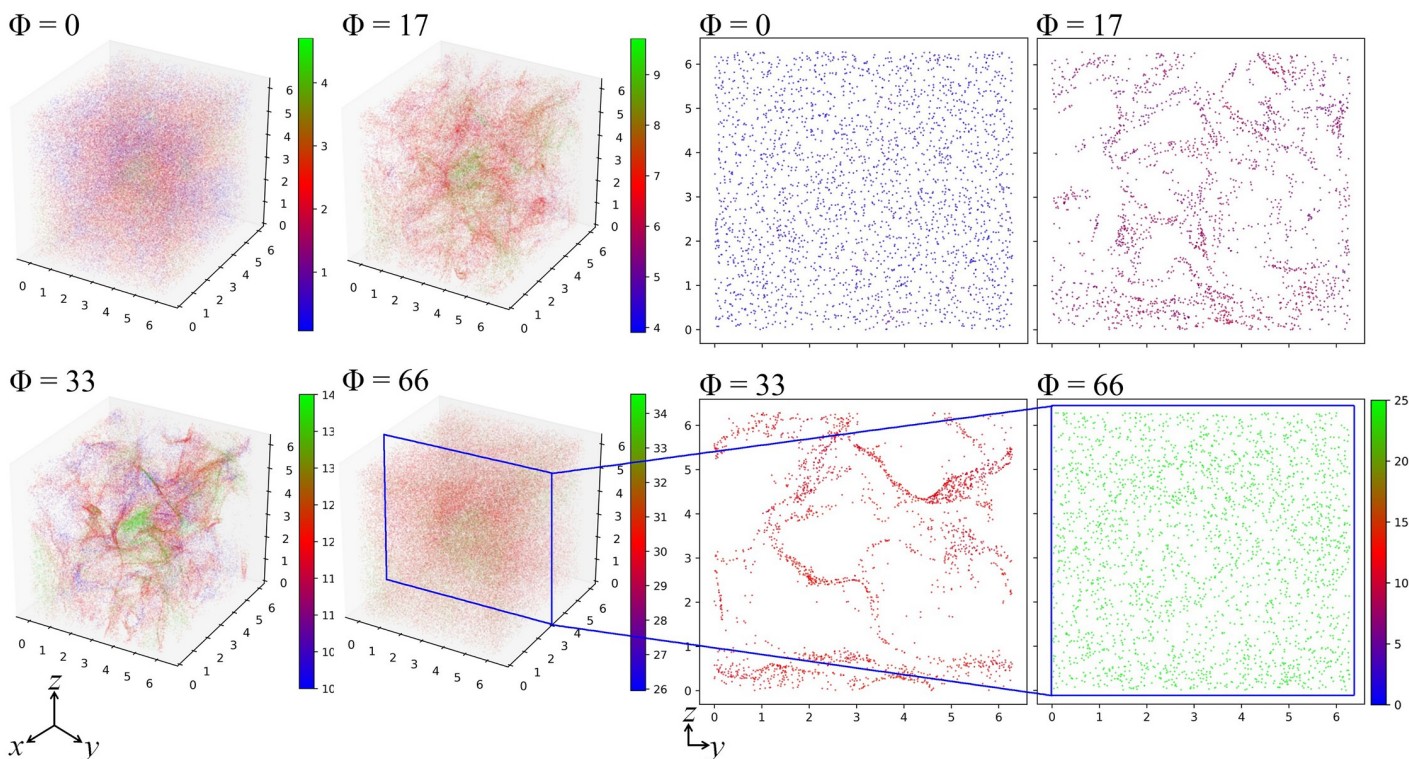

**Fig 2. Cluster formation of swimming particles.** Snapshots of the distribution of particles when $Re_\lambda = 59$ for swimming numbers $\Phi = [0, 17, 33, 66]$, where the swimming number $\Phi = 0$ corresponds to tracer particles. Colors correspond to the swimming speed of the particles.

values of $Re_\lambda$ where the results are included in the latter part of this section as Supporting information for a more organized presentation.

## Cluster formation

Particles seeded uniformly in the cube, with homogeneous and isotropic incompressible turbulent fluid flow, are allowed to disperse until their mass distribution reaches a statistically steady state. Fig 2 shows snapshots of the spatial distribution of particles in 2D and 3D for different swimming speeds. The 2D plot shows a slice across the $yz$-plane flattened along the $x$-axis with thickness $L_x/42$ where $L_x$ is the cube length along the $x$-axis. The colormap represents the swimming speed (from Eq 1) of the particles. Here we see clustering at swimming number $\Phi = 17$ and $\Phi = 33$ with the highest particle clustering observed at $\Phi = 33$ where relatively big voids in the 2D plot can be observed. However, at $\Phi = 66$, the particles become less accumulated again.

Fig 3A shows that the PDF of the particle density follows an exponential fit. The exponent $\lambda$ of the fit increases as more bins become populated with more particles. Particles with swimming number $\Phi = 33$, where we observed the highest clustering in Fig 2, have the highest $\lambda$. The trend for $\Phi = 66$, where we observed a decrease in clustering (in Fig 2), is comparable to $\Phi = 17$.

In our simulations, the lowest entropy $H$ is measured at $\Phi = 33$ while the highest $H$ is measured at $\Phi = 0$ (represented as a dashed line in Fig 3B). Correspondingly, swimming numbers $\Phi = 33$ and $\Phi = 0$ have the lowest and highest $\lambda$, respectively. The behavior of $H$ in our results has a similar trend with the previous results where clustering increases and reaches a

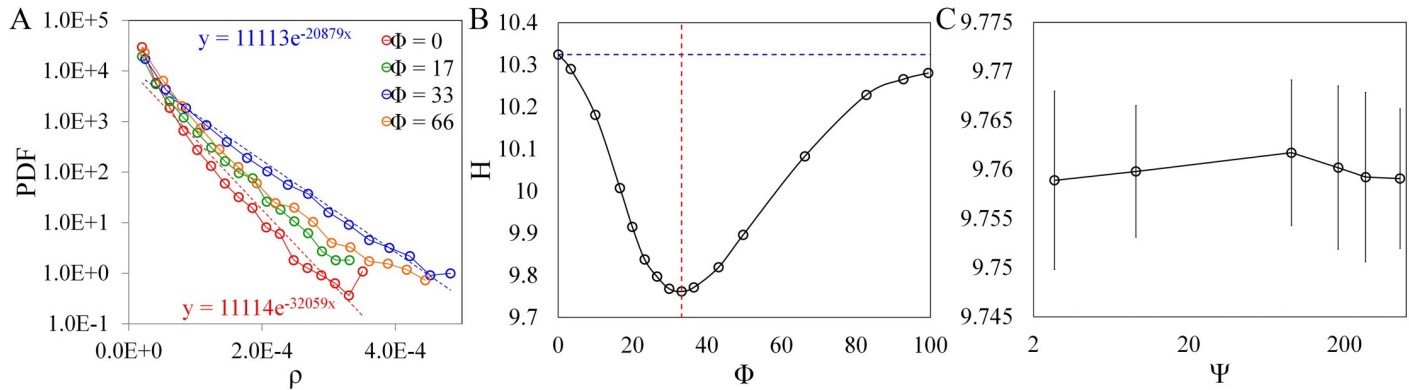

**Fig 3. Particle density PDF and entropy.** (A) Particle density PDF when $Re_\lambda = 59$ for different swimming numbers $\Phi = [0, 17, 33, 66]$. (B) Changes in entropy $H$ with swimming number at $\Psi = 91$ (vertical dashed line for $\Phi = 33$). (C) Entropy $H$ as a function of stability number at $\Phi = 33$. Error bars taken across 250-time steps.

maximum followed by a decreasing trend [9, 17]. However, in our results, a significant change in the clustering is observed when $\Phi$ is changed, while no significant difference in $H$ is observed when $\Psi$ is changed (see Fig 3C), which is in contrast with the previous studies where cluster formation is dominated by the stability number [9, 17]; cluster formation is only dictated by the swimming number. For other values of Reynolds number, the behavior of $H$ is similar to when $Re_\lambda = 59$ except the graphs shift towards higher $\Phi$ values (see S1 Fig). Consistent with the result when $Re_\lambda = 59$, clustering in the lower $Re_\lambda$ values is still invariant with the reorientation time (stability number) as shown in S1 Fig.

Our results show a cluster formation mechanism different from what has been previously reported. We deem that the difference in the observed clustering is attributed to the velocity and acceleration characteristics of the particles. Thus in the next sections, we explore the probability distributions of particle velocity and particle acceleration and compare them for different values of $\Phi$.

## Velocity and acceleration characteristics of swimming particles

To gain insights into the characteristics of particles as they are dispersed or clustered throughout the space, we look into the distribution of their velocity and acceleration. We focus our investigation on the effect of swimming number $\Phi$ on clustering.

**Velocity probability distribution.** In Fig 4A, we show the PDFs of the velocity magnitude $v = |\mathbf{v}|$ of the particles (from Eq 1) for different values of $\Phi$. Our result is consistent with previous results where PDF of velocity magnitude of Lagrangian(tracer) particles are observed to be nearly Gaussian [26, 27]. Here we observe a broadening in their distribution as $\Phi$ is increased. To quantify the broadening of the distribution, we take the Gaussian fit of the PDFs and compare the parameters $\sigma$ and $\mu$, the standard deviation, and the mean, respectively. In Fig 4B, we see a non-linear increase in $\sigma$ as $\Phi$ is increased, while in Fig 4C, we observe that increasing the swimming number $\Phi$ also increases the mean speed $\mu$. We expect these results since the particle velocity is just the superposition of both the swimming and fluid velocity as shown in Eq 1.

**Acceleration probability distribution.** Fig 5 shows the plots of the PDFs of the $x$-components of the three types of particle acceleration, namely, the acceleration $a$, centripetal acceleration $a_C$, and the longitudinal acceleration $a_L$, which are de-dimensionalized with the root-mean-squared acceleration $a_{rms}$. Our results shown in Fig 5A is in agreement with the previous studies [27, 28] where the tails of the distribution follow the exponential fit (represented as

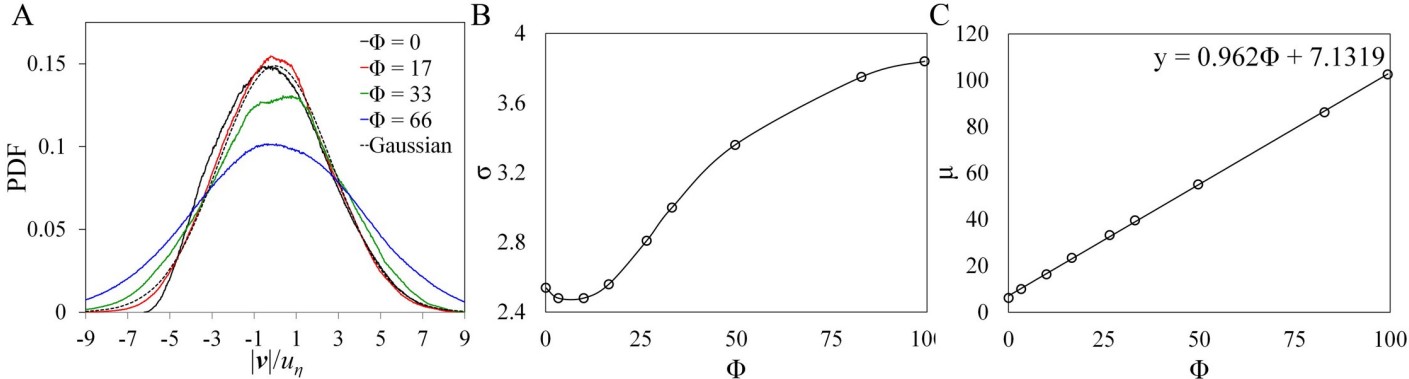

**Fig 4. PDF of microswimmer speed when $Re_\lambda = 59$.** All values are subtracted with the mean such that the distributions are centered at zero. (B) and (C) are plots of the standard deviation $\sigma$ and mean $\mu$ versus the swimming speed, respectively.

black dashed line) given by $P(a) = C \exp(-a^2/(1 + |a\beta/\gamma|^\delta)\gamma^2)$, where $\beta = 0.49$, $\gamma = 0.42$ and $\delta = 1.56$. Likewise, the plots for $a_C(x)$ and $a_L(x)$ have qualitatively comparable results with [29] where the tails of the longitudinal acceleration PDF drop more sharply compared to the centripetal acceleration PDF. Both PDFs for $a(x)$ and $a_C(x)$ have a wider range of values compared to the PDF of $a_L$, which ranges only up to $a_l(x)/a_{rms} < \pm 3$. The inset in Fig 5A, on the other hand, shows the plot of $a_{rms}$ vs.$\Phi$. The narrowing of the PDFs is a consequence of the increase in the $a_{rms}$ values as $\Phi$ is increased.

The PDFs of the magnitudes of $a$, $a_C$, and $a_L$ are shown in Fig 6. We also observe a narrowing of the distributions as $\Phi$ is increased, which is more pronounced at the high-acceleration tails. At low acceleration values, the PDFs follow a power-law fit (shown in Fig 6A–6C), which is represented as the solid black line. Except for a small difference in shifts, the PDFs follow the same trend with an average exponent of $1.93 \pm 0.01$. We observe a cross-over where the PDFs of swimming particles become greater compared to the PDFs of tracer particles ($\Phi = 0$). In Fig 6E and 6F, we show the log-linear representation of the plots where the high-acceleration tails are observed to be nearly exponential. The estimated exponential fit is shown as the solid black lines. Here we observe a second cross-over where tracer particles PDF becomes highest.

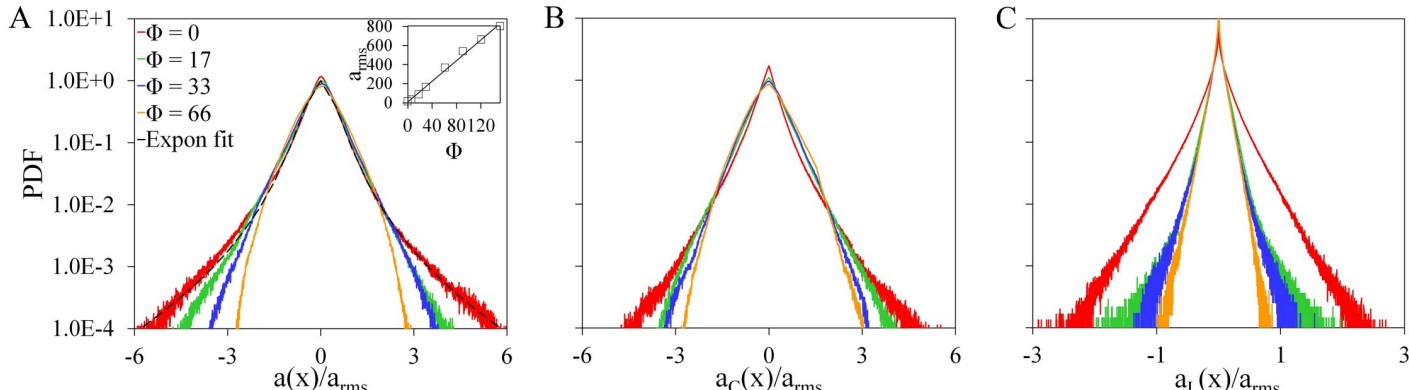

**Fig 5. PDF of the *x*-components of acceleration.** Shown are the PDFs of (A) acceleration $a(x)$, (B) centripetal $a_C(x)$, and (C) longitudinal acceleration $a_L(x)$ when $Re_\lambda = 59$ for $\Phi = 33$. Black dashed line in (A) corresponds to the exponential fit [27, 28] for $\Phi = 0$. Different colors correspond to swimming numbers $\Phi = [0, 17, 33, 66]$.

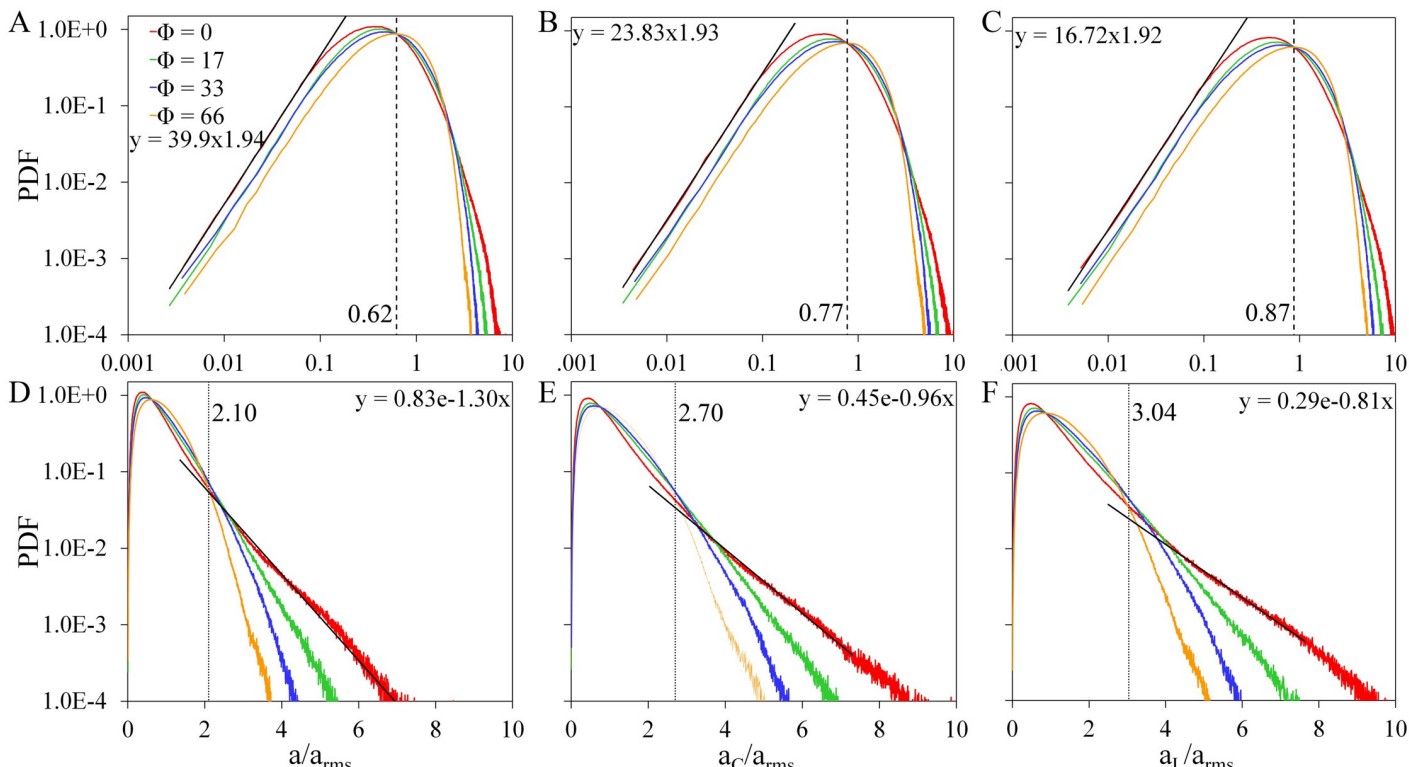

**Fig 6. PDF of acceleration magnitude when $Re_\lambda = 59$.** (A-C) Log-log plots of the acceleration magnitude PDF where the vertical dashed line corresponds to the location of the first cross-over. (D-F) Log-linear plots of the the PDFs where the vertical dotted lines corresponds to the location of the second cross-over. Black solid lines are the estimated fits.

We observe a narrowing in the PDFs, which is attributed to the increase in $a_{rms}$. We also observed two kinds of deviations from the distributions of tracer particles ($\Phi = 0$), i.e., (a) the first cross-over where the PDFs of swimming particles become higher and (b) the second cross-over where the PDF of tracer particles becomes the highest. The two cross-overs observed in the PDFs may have a contribution to the clustering of swimming particles. To investigate further, we look into the acceleration characteristics of the most clustered particles or particles in bins with density $\rho \geq 0.00015$. We take the average acceleration of the particles contained in each bin and plot the frequency distribution as shown in Fig 7. For all types of accelerations, we observe the highest bin counts at $\Phi = 33$ where clustering is maximum, and the lowest counts for $\Phi = 0$ where no clustering is observed. Peaks in the frequency distributions (shown in Fig 7A–7C) are observed at acceleration values where the first cross-over in Fig 6 is observed, while the peaks in the plots for accelerations along the $x$-axis (shown in Fig 7D–7F) are at 0. We can see that the most clustered particles reside within acceleration values near the first cross-over location, and the number of bins becomes zero as it approaches the second cross-over location. For all the other values of $Re_\lambda$, we also observed two cross-overs in their corresponding acceleration PDFs with a slight change in locations. Consistent with the results when $Re_\lambda = 59$, the peaks in the frequency distributions of the most clustered particles are also observed at the first cross-over location when $Re_\lambda = [21, 36]$ as shown in S2 and S3 Figs. This means that the presence of the first cross-over is an indicator that acceleration drives the clustering of swimming particles described here. On the other hand, there is no significant difference in the frequency distribution plots except for the increased peak in the plot for $a_L(x)$

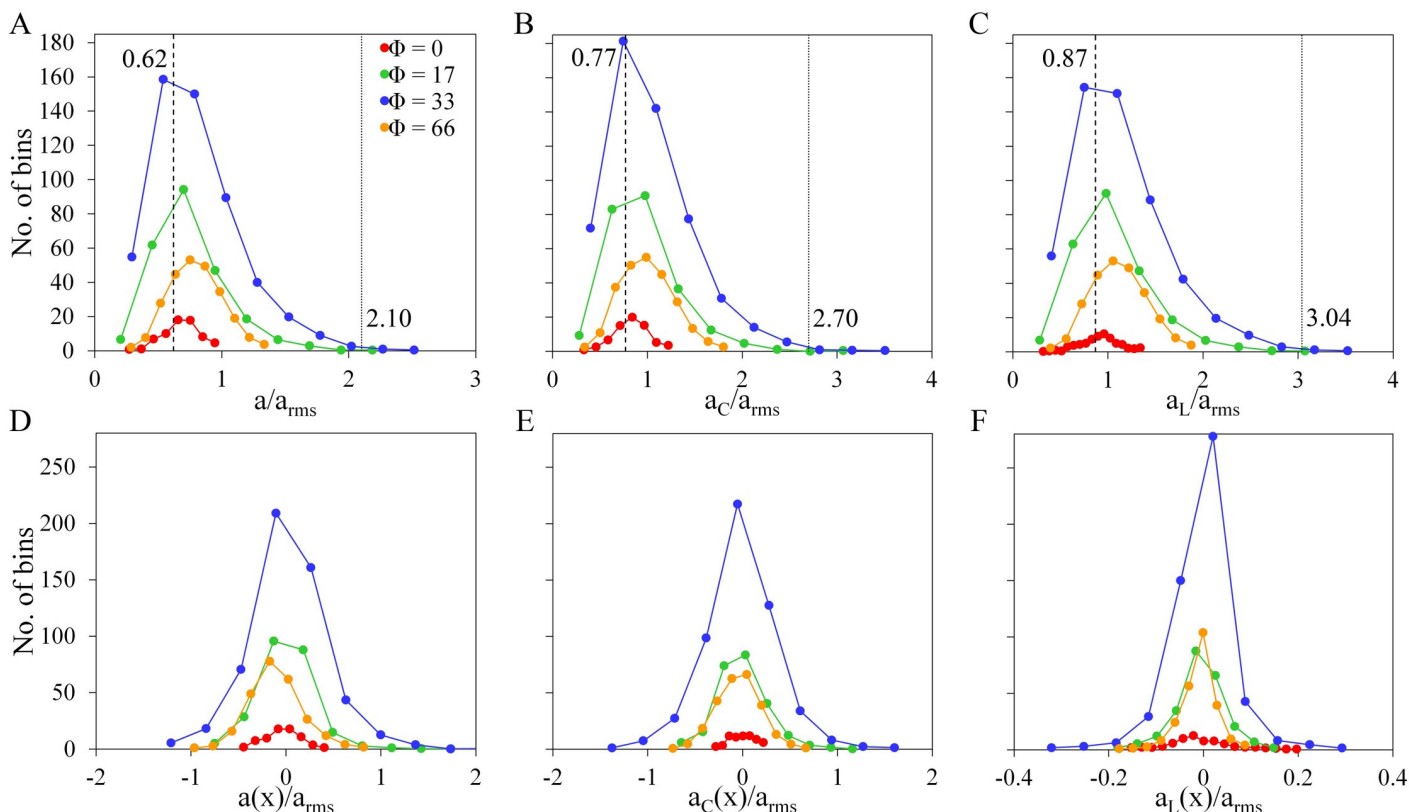

**Fig 7. Histogram of the magnitudes and *x*-components of acceleration when *Re*$_\lambda$ = 59.** The number of bins (cube) per acceleration value for the magnitudes of (A) acceleration, (B) centripetal acceleration, and (C) longitudinal acceleration. (D-F) corresponds to the *x*-components of the three accelerations.

(Fig 7F). Thus, the cluster formation is not dictated by vortex trapping. Our results on the mechanism of cluster formation can be applied to further understand the dynamics of natural/ artificial fast swimmers. While clustering is advantageous during reproduction as it enhances the encounter rates between cells [9], it can also be detrimental as it increases the transmission of diseases. With our results, we may be able to gain insights into the balance between cluster- ing and transmission of diseases.

## Conclusions

Previous studies have shown that clustering of microswimmers is greatly affected by both $\Phi$ and $\Psi$, which results in particles accumulating inside vortices and in downwelling (downward flow velocity) regions [9, 11, 12]. Clustering in the downwelling regions occurs specifically for particles in the limit of $\Psi \ll 1$. In our simulations, the clustering is only significantly affected by swimming speed and remains invariant for any $\Psi$. Here, the swimming numbers explored in our simulations are much higher than the previous reports. Our results show that clustering can be achieved even without the influence of the stabilizing torque as long as the particles swim at high speeds. The highest particle clustering was achieved at moderate values of swim- ming number. As $\Phi$ increases, the particles move faster and become more dispersed resulting in a decrease in clustering. Furthermore, clusters emerged from balancing the effects of motil- ity and vorticity, which has been previously discussed in the references [9–11]. The turbulent flow in our system is in a low Reynolds number, $Re_\lambda$ = [21, 36, 59], such that shear from

vortices cannot form clusters, but by adding motility to the particles, clusters can start to form. However, adding a very high swimming velocity to the particles breaks the balance, which then results in a decrease in $H$. The mechanism of cluster formation presented in this paper can be applied to understand many natural/artificial swimmers with faster swimming speeds [1, 2] where clustering can be advantageous for reproduction as it enhances the encounter rates between cells, but it can also be detrimental as it increases the transmission of diseases.

Our results may also provide useful insights into water system management since most urban water contamination is caused by motile bacteria. We have previously noted that the presence of intense and persistent centripetal acceleration is correlated with intense vorticity, which is the evidence of particles trapped inside vortices [16]. In our simulations, the majority of the particles in high-density bins have small magnitudes of acceleration indicating that clustered particles are located outside regions of high vorticity. The observed clustering is similar to that of clustering from vortex ejection, which to our knowledge has not yet been observed in any work on clustering of motile microorganisms. In addition, the acceleration of most of the clustered particles resides within the cross-over location in the acceleration PDFs, where the PDFs of swimming particles become higher compared to that of tracer particles. Our results reveal that particle acceleration drives clustering.

## Supporting information

**S1 Table. Simulation parameters used in different Reynolds number flow.** Swimming parameters and number of particles used in each simulation for different values of Reynolds number.
(TIF)

**S1 Fig. Entropy $H$ versus swimming number $\Phi$ and reorientation time $B$.** Changes in the entropy $H$ with (A) swimming number $\Phi$, and (B) reorientation time $B$ for different values of Reynolds number $Re_\lambda$ at $\Phi = 33$.
(TIF)

**S2 Fig. The number of bins (cube) per acceleration value (magnitude and $x$-components) for $Re_\lambda = 21$.** Histogram for (A) acceleration, (B) centripetal acceleration, and (C) longitudinal acceleration while (D-F) corresponds to the histograms for the $x$-components of the three accelerations.
(TIF)

**S3 Fig. The number of bins (cube) per acceleration value (magnitude and $x$-components) for $Re_\lambda = 36$.** Histogram for (A) acceleration, (B) centripetal acceleration, and (C) longitudinal acceleration while (D-F) corresponds to the histograms for the $x$-components of the three accelerations.
(TIF)

## Acknowledgments

The authors woud like to thank Professors Danilo M. Yanga, Christopher P. Monterola, and Christian M. Alis for the fruitful discussions about this work.

## Author Contributions

**Conceptualization:** Jenny Lynn Ongue Almerol, Marissa Pastor Liponhay.

**Formal analysis:** Jenny Lynn Ongue Almerol, Marissa Pastor Liponhay.

**Investigation:** Jenny Lynn Ongue Almerol.

**Methodology:** Jenny Lynn Ongue Almerol.

**Software:** Jenny Lynn Ongue Almerol.

**Supervision:** Marissa Pastor Liponhay.

**Validation:** Jenny Lynn Ongue Almerol, Marissa Pastor Liponhay.

**Visualization:** Jenny Lynn Ongue Almerol.

**Writing – original draft:** Jenny Lynn Ongue Almerol.

**Writing – review & editing:** Jenny Lynn Ongue Almerol, Marissa Pastor Liponhay.

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
