## [Decision Letter · Decision Letter 0]

10 Jan 2022

PONE-D-21-37882Clustering of fast gyrotactic particles in low-Reynolds-number flowPLOS ONE

Dear Dr. Almerol,

Thank you for submitting your manuscript to PLOS ONE. After careful consideration, we feel that it has merit but does not fully meet PLOS ONE’s publication criteria as it currently stands. Therefore, we invite you to submit a revised version of the manuscript that addresses the points raised during the review process.

We look forward to receiving your revised manuscript.

Kind regards,

Fang-Bao Tian

Academic Editor

PLOS ONE

Journal Requirements:

[We would like to thank the Department of Science and Technology (DOST)-SEI 252

Accelerated Science and Technology Human Resource Development Program (ASTHRDP) and University of San Carlos for supporting this research. M. Liponhay would also like to acknowledge the funding from DOST-CRADLE Program, Project No. 8419. Finally, we thank Professors Danilo M. Yanga, Christopher P. Monterola, and Christian M. Alis for the fruitful discussions about this work.]

 [The author(s) received no specific funding for this work.]

Reviewers' comments:

Reviewer's Responses to Questions

**Comments to the Author**

1. Is the manuscript technically sound, and do the data support the conclusions?

Reviewer #1: No

2. Has the statistical analysis been performed appropriately and rigorously? 

Reviewer #1: Yes

3. Have the authors made all data underlying the findings in their manuscript fully available?

Reviewer #1: Yes

4. Is the manuscript presented in an intelligible fashion and written in standard English?

Reviewer #1: Yes

5. Review Comments to the Author

Reviewer #1: This manuscript is on the clustering of fast gyro-tactic particles in low-Reynolds-number flow, which is beneficial to understand various ecological phenomena, for example, harmful algal blooms in lakes. The authors' efforts are appreciated, and the authors should consider the following points to improve further the quality of this manuscript.

(1) The primary method used in this paper was reported in previous work by Durham et al. (Durham WM, Climent E, Barry M, De Lillo F, Boffetta G, Cencini M, et al. Turbulence drives microscale patches of motile phytoplankton. Nature communications. 2013). This work presents swimming particles' velocity and acceleration characteristics for different parameters. However, the numerical analysis in this work has not been verified by some experiments and relevant literature. At least, more cases of numerical simulation should be performed and analyzed for flows of various Reynolds numbers to make the conclusion powerful.

(2) The conclusion is given for fast gyrotactic particles in low-Reynolds-number flow. More limitations are required to specify the scope of the conclusion, for example, the range of low-Reynolds-number flow.

6. PLOS authors have the option to publish the peer review history of their article (what does this mean?). If published, this will include your full peer review and any attached files.

Reviewer #1: No

---

## [Author Response · Author response to Decision Letter 0]

27 Jan 2022

RESPONSE TO THE EDITOR 

COMMENT 1: Please ensure that your manuscript meets PLOS ONE's style requirements, including those for file naming. The PLOS ONE style templates can be found at 

RESPONSE 1: Thank you for the resources. We have

 Added corresponding author’s initials beside email address.

 Removed funding information from the Acknowledgements section. 

 Improved the citations of equations and figures in Methods and Results and discussions sections. 

COMMENT 2: Thank you for stating the following in the Acknowledgments Section of your manuscript: 

[We would like to thank the Department of Science and Technology (DOST)-SEI 252

Accelerated Science and Technology Human Resource Development Program (ASTHRDP) and University of San Carlos for supporting this research. M. Liponhay would also like to acknowledge the funding from DOST-CRADLE Program, Project No. 8419. Finally, we thank Professors Danilo M. Yanga, Christopher P. Monterola, and Christian M. Alis for the fruitful discussions about this work.]

 [The author(s) received no specific funding for this work.]

RESPONSE 2: Concerning the Acknowledgments Section of the manuscript, we have removed the funding-related statement and incorporated this correction in the revised manuscript. We propose the following amendment to the funding statement:

“JA and ML acknowledge the Department of Science and Technology (DOST)-SEI Accelerated Science and Technology Human Resource Development Program (ASTHRDP) and the University of San Carlos for supporting this research. ML acknowledges the Department of Science and Technology (DOST) of the Philippines with Project No. 8419, 2020 under the Collaborative R&D to Leverage the Economy (CRADLE) Program (https://s4cp.dost.gov.ph/programs/cradle/). The funders had no role in study design, data collection, and analysis, decision to publish, or preparation of the manuscript.”

COMMENT 3: We note that you have stated that you will provide repository information for your data at acceptance. Should your manuscript be accepted for publication, we will hold it until you provide the relevant accession numbers or DOIs necessary to access your data. If you wish to make changes to your Data Availability statement, please describe these changes in your cover letter and we will update your Data Availability statement to reflect the information you provide.

RESPONSE 3: Thank you for the information. We are changing our Data Availability statement to

“All preprocessed data files are available from 10.5281/zenodo.5904617, 10.5281/zenodo.5905360, and 10.5281/zenodo.5904923.”

We have also mentioned this in our cover letter.

RESPONSE TO THE REVIEWER 

Reviewer #1: This manuscript is on the clustering of fast gyro-tactic particles in low-Reynolds-number flow, which is beneficial to understand various ecological phenomena, for example, harmful algal blooms in lakes. The authors' efforts are appreciated, and the authors should consider the following points to improve further the quality of this manuscript.

COMMENT 1: The primary method used in this paper was reported in previous work by Durham et al. (Durham WM, Climent E, Barry M, De Lillo F, Boffetta G, Cencini M, et al. Turbulence drives microscale patches of motile phytoplankton. Nature communications. 2013). This work presents swimming particles' velocity and acceleration characteristics for different parameters. However, the numerical analysis in this work has not been verified by some experiments and relevant literature. At least, more cases of numerical simulation should be performed and analyzed for flows of various Reynolds numbers to make the conclusion powerful.

RESPONSE 1: We thank the reviewer for this comment. We have performed additional numerical simulations using different Reynolds numbers, where clustering of gyrotatic cells is observed inside the vortices in the previous studies. We use similar values to confirm whether the mechanism of particle clustering discussed in our study can be observed in such flows. 

We found the same behavior for Re_λ=[21,36] – the presence of clustering with the same characteristics with respect to velocity and acceleration distribution as when Re_λ=59, which supports our conclusions. Shown below are the corresponding snapshots of the distribution of particles for Re_λ=[21,36] for different swimming numbers.

We have also provided additional discussions in the Results and discussions section (in the third paragraph of the Cluster formation subsection and in the last paragraph of the Velocity and acceleration characteristics of swimming particles subsection) to support our conclusions and have incorporated the results in the manuscript as supporting information.

COMMENT 2: The conclusion is given for fast gyrotactic particles in low-Reynolds-number flow. More limitations are required to specify the scope of the conclusion, for example, the range of low-Reynolds-number flow.

RESPONSE 2: We thank the reviewer for the suggestion. We agree with the reviewer, and we have included the range of low-Reynolds-number flow and the basis of selection of these values in Paragraph 2 of the Numerical procedure subsection of Methods. 

Our findings that motile particles cluster outside vortices when also happen for low Re numbers, i.e., when the particle speed is high enough. Given these observations, our data and findings now better support the conclusions with its scope for fluid flow at low Reynolds number and high particle speed. The results of the numerical experiments using these values have been discussed in the Results and Discussions and are presented in the Supporting information section.

In addition to the revisions made to the manuscript above, we have also made corrections in the values used to de-dimensionalize the swimming parameters, changing the values of the swimming parameters. These changes only affect the swimming parameter values reported in the figures (axis values in Figs 3 and 4 and legends of Figs 2-7) and do not affect the trend in our plots or the analyses of the data. We have incorporated these corrections in the Methodology and Results and Discussions sections of the manuscript.

We also added a new paragraph at the beginning of the Results and discussion section to add clarity in the organization of the presentation of results.

---

## [Decision Letter · Decision Letter 1]

8 Mar 2022

PONE-D-21-37882R1Clustering of fast gyrotactic particles in low-Reynolds-number flowPLOS ONE

Dear Dr. Almerol,

Thank you for submitting your manuscript to PLOS ONE. After careful consideration, we feel that it has merit but does not fully meet PLOS ONE’s publication criteria as it currently stands. Therefore, we invite you to submit a revised version of the manuscript that addresses the points raised during the review process.

We look forward to receiving your revised manuscript.

Kind regards,

Fang-Bao Tian

Academic Editor

PLOS ONE

Journal Requirements:

Reviewers' comments:

Reviewer's Responses to Questions

**Comments to the Author**

1. If the authors have adequately addressed your comments raised in a previous round of review and you feel that this manuscript is now acceptable for publication, you may indicate that here to bypass the “Comments to the Author” section, enter your conflict of interest statement in the “Confidential to Editor” section, and submit your "Accept" recommendation.

Reviewer #1: (No Response)

2. Is the manuscript technically sound, and do the data support the conclusions?

Reviewer #1: Yes

3. Has the statistical analysis been performed appropriately and rigorously? 

Reviewer #1: Yes

4. Have the authors made all data underlying the findings in their manuscript fully available?

Reviewer #1: Yes

5. Is the manuscript presented in an intelligible fashion and written in standard English?

Reviewer #1: Yes

6. Review Comments to the Author

Reviewer #1: (1) Please check Eq.2; it is not written correctly. p->dp/dt

(2) The references for Eqs. 1 and 2 should be given.

(3) Figures are unclear, and authors should make sure all figures are clear enough for reading.

(4) Texts and languages should be improved further. For example, Navier-Stokes NS equation -> Navier-Stokes equation.

7. PLOS authors have the option to publish the peer review history of their article (what does this mean?). If published, this will include your full peer review and any attached files.

Reviewer #1: No

---

## [Author Response · Author response to Decision Letter 1]

16 Mar 2022

RESPONSE TO THE EDITOR

Journal Requirements:

Response:

We thank the editor for this comment. We have checked our reference list and found no articles that have been retracted. We made no major changes in the reference list except for a few additional information added in reference numbers 8, 21, 24, and 25. 

RESPONSE TO THE REVIEWER 

Reviewer #1: 

(1) Please check Eq.2; it is not written correctly. p->dp/dt

Thank you for spotting this error. We have made corrections to the expression of Eq. 2 and have incorporated it in the revised manuscript.

(2) The references for Eqs. 1 and 2 should be given.

We thank the reviewer for this comment. We have now cited the reference for Eqs. 1 and 2 in line number 66. We have also checked if there are parts in the manuscript where we may have failed to properly cite the references and made the necessary corrections in the revised manuscript. We cited the references in line numbers 29-30 and line numbers 120-122.

(3) Figures are unclear, and authors should make sure all figures are clear enough for reading.

Thank you for pointing out this concern. We have improved our figures and increased their resolutions making sure that they are all clear enough for reading. We have also revised Fig 7, S3 Fig and S4 Fig so that the axis titles are shown.

(4) Texts and languages should be improved further. For example, Navier-Stokes NS equation -> Navier-Stokes equation.

Regarding this concern, we have now improved the texts and languages used in the manuscript. 

We have changed the Navier-Stokes NS equation to Navier-Stokes (NS) equation, as NS equation is repeatedly used in the manuscript. Similar necessary changes have also been made throughout the manuscript to remove inconsistencies in the usage of other terms and abbreviations. 

We have also significantly improved the clarity of the discussions in our manuscript by revisiting our choices of words and by polishing our use of the English language.

---

## [Editor Report · Decision Letter 2]

24 Mar 2022

Clustering of fast gyrotactic particles in low-Reynolds-number flow

PONE-D-21-37882R2

Dear Dr. Almerol,

We’re pleased to inform you that your manuscript has been judged scientifically suitable for publication and will be formally accepted for publication once it meets all outstanding technical requirements.

Kind regards,

Fang-Bao Tian

Academic Editor

PLOS ONE
---

## [Editor Report · Acceptance letter]

30 Mar 2022

PONE-D-21-37882R2 

Clustering of fast gyrotactic particles in low-Reynolds-number flow 

Dear Dr. Almerol:

I'm pleased to inform you that your manuscript has been deemed suitable for publication in PLOS ONE. Congratulations! Your manuscript is now with our production department. 

Kind regards, 

on behalf of

Prof. Fang-Bao Tian 

Academic Editor

PLOS ONE